# Smoke Suppression in Electron Beam Melting of Inconel 718 Alloy Powder Based on Insulator–Metal Transition of Surface Oxide Film by Mechanical Stimulation

**DOI:** 10.3390/ma14164662

**Published:** 2021-08-18

**Authors:** Akihiko Chiba, Yohei Daino, Kenta Aoyagi, Kenta Yamanaka

**Affiliations:** Institute for Materials Research, Tohoku University, 2-1-1 Katahira, Aoba-ku, Sendai 980-8577, Japan; ydaino@jeol.co.jp (Y.D.); k.aoyagi@imr.tohoku.ac.jp (K.A.); k_yamanaka@imr.tohoku.ac.jp (K.Y.)

**Keywords:** additive manufacturing, electron beam melting, powder bed fusion, smoking, ball milling, oxide film, Inconel 718, electrical resistivity, metal–insulator transition, Mott insulator

## Abstract

In powder bed fusion–electron beam melting, the alloy powder can scatter under electron beam irradiation. When this phenomenon—known as smoking—occurs, it makes the PBF-EBM process almost impossible. Therefore, avoiding smoking in EBM is an important research issue. In this study, we aimed to clarify the effects of powder bed preheating and mechanical stimulation on the suppression of smoking in the powder bed fusion–electron beam melting process. Direct current electrical resistivity and alternating current impedance spectroscopy measurements were conducted on Inconel 718 alloy powder at room temperature and elevated temperatures before and after mechanical stimulation (ball milling for 10–60 min) to investigate changes in the electrical properties of the surface oxide film, alongside X-ray photoelectron spectroscopy to identify the surface chemical composition. Smoking tests confirmed that preheating and ball milling both suppressed smoking. Furthermore, smoking did not occur after ball milling, even when the powder bed was not preheated. This is because the oxide film undergoes a dielectric–metallic transition due to the lattice strain introduced by ball milling. Our results are expected to benefit the development of the powder bed fusion–electron beam melting processes from the perspective of materials technology and optimization of the process conditions and powder properties to suppress smoking.

## 1. Introduction

Electron beam melting (EBM) is an additive manufacturing process that produces components by selectively melting layers of metal powder based on three-dimensional computer-aided design data [1,2,3,4,5]. Compared to laser beam additive manufacturing processes, EBM has many advantages [2] with regard to the processing speed, energy density, residual stresses, and applicability to metal powders with high melting temperatures such as Ta [6]. However, the electron beam has a negative charge, which introduces several complexities with regard to the control of the EBM process. For instance, the “smoking” phenomenon [7,8,9,10] is an intrinsic problem in powder bed fusion (PBF)-EBM. Metal powder particles are negatively charged when the powder bed is subjected to electron beam irradiation, which means they scatter “like smoke” when the Coulomb force exceeds a critical value [7,8]. When smoking occurs, the building process becomes virtually impossible. Therefore, this problem must be overcome to expand the application scope of PBF-EBM processes.

Several methods have been adopted so far to suppress smoking. The prevalent technique is to preheat the powder bed prior to the melting process, which is effective for removing the residual stresses that accumulate during layer-by-layer melting. Generally, if smoking does not occur during the preheating process, then it will not occur during the melting process. However, if the temperature used for preheating is too high, it can cause solid-state sintering of the powder bed, which leads to difficulties in removing the unmelted powder from the built components. Thus, the preheating temperature for a particular powder is normally determined by considering the lowest temperature at which smoking does not occur. In addition, coarse powders are generally less likely to smoke than fine powders. However, these empirical methods of suppressing smoking lack a physical explanation and are not always useful in predicting whether a given building parameter will suppress smoking. Therefore, a basic understanding of the charging behavior of alloy powders is important for accurately predicting the building parameters that cause smoking. Although this is an extremely important research area for the development of PBF-EBM technology, the current research and development from this perspective is scarce.

Most alloy powders are covered by a surface oxide film with a thickness of several nanometers. For example, powders of Inconel 718 containing 10% or more Cr [11] and Co–28Cr–6Mo for medical use [12] form oxide films containing Cr_2_O_3_ as the main component. Furthermore, powders of titanium alloys, such as Ti–6Al–4V, form an oxide film containing TiO_2_ as the main component. Oxides of 3d transition metals, such as Cr and Ti, are considered to be Mott insulators, which behave as electrical insulators [13]. Therefore, because neighboring powder particles are in contact via the insulating oxide film, the electrical resistance of these powders would be as large as that of an insulator. When subjected to electron beam irradiation, the electrons can penetrate through the surface oxide film and into the alloy to a depth of approximately 10 μm. For the interior of the powder particles to remain electrically neutral, the penetrating electrons must flow to the adjacent alloy powder particles. However, the insulating surface oxide film can impede conduction between adjacent particles.

At present, it is unclear how the electrons accumulate in the oxide film; however, the oxide film certainly plays an important role in the accumulation of negative charge in alloy powders that are subjected to electron beam irradiation. Because the surface oxide film is extremely thin, the chemical composition may easily change upon heating. Consequently, the electrical resistivity of the powder is likely to decrease as the temperature is increased, which may explain why preheating suppresses smoking. Therefore, the mechanism of smoke generation upon electron beam irradiation can be clarified by elucidating the electrical properties of the surface oxide film, and how they change upon preheating. Furthermore, if the electrical characteristics of the surface oxide film could be changed by a method other than preheating, it can be inferred that smoke generation could be controlled without preheating.

Accurately measuring the temperature dependence of the electrical resistivity of the alloy powder between room temperature and approximately 1000 °C is of great importance in determining the relationship between the preheating temperature and smoke generation behavior during PBF-EBM. Useful findings can be obtained as an evaluation of thermal stability. In addition, alternating current (AC) impedance measurements are effective for investigating the electrical characteristics of the surface oxide film. This is because AC impedance measurements can separate the capacitance (dielectric), resistance, and inductive components of the electrical characteristics. It is particularly important to elucidate the electrical properties of the surface oxide film, rather than the absolute electrical resistivity of the alloy. Therefore, by studying how the electrical properties of the surface oxide film change with temperature, the empirical fact that smoking can be suppressed by preheating the powder bed can be quantitatively understood.

Furthermore, we hypothesize that, if the surface oxide film is a Mott insulator, an insulator/metal transition can be induced by introducing uniaxial lattice strain into the oxide film, such as by mechanical stimulation (e.g., by ball milling) [13]. If it could be metallically conveyed to the surface oxide film, the accumulation of negative charge would be suppressed, thereby avoiding smoke generation. To explore this hypothesis, the changes in the electrical properties of the alloy powder due to mechanical stimulation should be investigated, such as by measuring the electrical properties of the powder before and after ball milling.

The purpose of this study was to clarify the effect of powder bed preheating on smoke formation in the PBF-EBM process. We focused on Inconel 718 alloy powder, whose surface is covered with an insulating Cr_2_O_3_ passivation film with a thickness of several nanometers [11], and investigated the temperature dependence of the electrical properties, such as the electrical resistivity and AC impedance, and the thermal stability of the surface oxide film. In addition, ball milling of the Inconel 718 alloy powder was conducted followed by a series of electrical measurements. Using this method, we investigated the effect of mechanical stimulation on the electrical properties of the surface oxide film. Finally, by performing smoking tests, we verified the effect of preheating and mechanical stimulation (ball milling) on the generation of smoke.

## 2. Materials and Methods

### 2.1. Materials

The powder used in this study was Inconel 718 powder fabricated via plasma atomization (hereafter designated by PA powder), which was purchased from Arcam AB (Mölndal, Sweden). Inconel 718 is a nickel-based superalloy that is commonly used as a high-temperature structural material in additive manufacturing in the aerospace and energy industries [14]. The chemical composition of the powder, as shown in Table 1, was measured by inductively coupled plasma atomic emission spectroscopy (IRIS Advantage DUO, Thermo Jarrell Ash Corp., Franklin, MA, USA). The particle size distribution of the Inconel 718 alloy powder was measured using a laser diffraction particle size analyzer (LS230, Beckman Coulter, Inc., Brea, CA, USA). Figure 1a shows a scanning electron microscopy (SEM) image of the PA powder used in this study. Although some of the powder particles had satellite particles attached, most powder particles were nearly spherical. Figure 1b,c show SEM images of the powder surface at higher magnifications, which demonstrate that the powder particles had a dendritic structure.

### 2.2. Mechanical Ball Milling

In this study, the purpose of ball milling was to introduce plastic strain into the oxide film on the powder surface, not to remove the surface oxide film. Therefore, air was selected as the processing atmosphere for ball milling. The ball milling treatment was performed in air using a high-energy planetary ball mill (Pulverisette 7, Fritsch GmbH, Idar-Oberstein, Germany) with tungsten carbide (WC) balls (diameter of 6 mm) as the milling media. The alloy powder and WC balls were placed into a stainless steel vessel at a volume ratio of 1:2. Processing times of 10, 30, and 60 min were used, wherein the rotation was carried out in a clockwise direction for the first half of the time, and in a counterclockwise direction for the second half. The rotation speed was 800 min^−1^. 

### 2.3. XPS Analysis

Surface chemical analysis of the untreated (virgin) and ball-milled (30 min) Inconel 718 alloy PA powders was performed using X-ray photoelectron spectroscopy (XPS, Kratos Analytical Ltd., Manchester, UK) with a monochromatic Al Kα source (1486.7 eV). The powder samples were prepared by distributing a thin layer of powder on carbon tape. Survey spectra from the powder surfaces were recorded to identify the presence of O 1s, Ti 2p, Al 2p, Cr 2p, Mo 3d, Ni 2p, and Nb 3d peaks. The XPS survey spectra were analyzed to determine the chemical composition of the powder surface using COMPRO12 software. The strong C 1s peak with a binding energy of 284.8 eV was adopted as a calibration standard for correcting the peak positions of the other elements. 

### 2.4. Electrical Resistivity and AC Impedance Measurements

The direct current (DC) electrical resistivity of the powders was measured by DC four-point probe electrical resistivity measurements. In addition, assuming the surface oxide film has dielectric (electrically insulating) characteristics, the powder bed can be regarded as a composite material composed of metal and dielectric components; thus, AC impedance spectroscopy was carried out on the virgin and ball-milled powders to evaluate the electrical properties, including the resistance and capacitance. For the electrical resistivity and AC impedance measurements, the powder sample was placed in an alumina tube with an inner diameter of 10 mm and a height of 30 mm, and the top and bottom sides of the powder were sandwiched between nickel electrodes. The DC electrical resistivity measurements were conducted in vacuum between room temperature (23 °C) and 800 °C using a DC voltmeter. The samples were heated from room temperature to 800 °C at a heating rate of 5 °C s^−1^; held for 1 h at 800 °C; and then cooled to room temperature at a cooling rate of 5 °C s^−1^. The AC impedance measurements were conducted using an inductance (*L*), capacitance (*C*), and resistance (*R*) (LCR) meter (ZM 2376 LCR, NF Corp., Yokohama, Japan). The measurements were performed in vacuum at room temperature and at 50, 100, 200, 300, 400, and 800 °C; voltages of 0.01 to 1 V and frequencies of 10 to 1.83 MHz were used. The software package EC-LAB (BioLogic Science Instruments Ltd., Grenoble, France) was used to simulate the output data of the impedance tests.

### 2.5. Smoke Detection and Hatching Experiments

“Smoking” experiments were conducted using a PBF-EBM machine developed by the Technology Research Association for Future Additive Manufacturing (TRAFAM) to examine the effect of preheating and powder surface modification on the smoking behavior. The experiments were performed at an accelerating voltage of 60 kV under a vacuum of 10^−2^ Pa. A schematic of the experimental equipment used for the smoking tests is shown in Figure 2. A 10 mm × 10 mm × 1.0 mm groove was formed in the center of a 100 mm × 100 mm × 10 mm stainless steel plate, and the groove was filled with powder. During the smoking tests, the stainless steel plate was first heated using an electron beam, avoiding the powder bed, to achieve preheating of an arbitrary temperature. The temperature of the stainless steel plate was measured using a thermocouple installed on the back surface of the stainless steel plate. Before the electron beam was focused onto the powder, it was intentionally deflected in the *X*-direction, and the signal was captured and triggered. The deflection condenser lens signal was captured using an oscilloscope to accurately measure the beam irradiation time and smoke generation time. The output of the oscilloscope was used to trigger the filming start time of a high-speed camera set up at the observation window on the front of the vacuum chamber of the PBF-EBM machine. The high-speed camera was used to determine whether smoke was generated during the smoking test.

During the smoking tests, pulse beam irradiation was used to more closely simulate the actual experimental characteristics. The beam irradiation patterns are shown in Figure 3. An electron beam with current *I* was irradiated for a time of 1/*f*_2_ at a frequency of *f*_1_. *f*_1_, *f*_2_, *I* were adjusted arbitrarily, and the presence or absence of smoking and the time until smoking were measured to elucidate the mechanism of smoke generation. Table 2 lists the experimental conditions for the smoke tests.

Hatching (powder bed melting) experiments were conducted to simulate the actual experimental conditions. A 10 mm × 10 mm × 0.2 mm groove in the center of a 100 mm × 100 mm × 10 mm stainless steel plate was filled with ball-milled powder. The base plate was preheated to 500 °C by electron beam irradiation, avoiding the powder bed. Ordinarily, the PBF-EBAM process of Inconel 718 alloy powder uses a preheating temperature of approximately 1050 °C; therefore, a preheating temperature of 500 °C is below the smoking temperature of the virgin powder. Subsequently, additional heat was applied to the powder bed by electron beam irradiation for 24 s with a beam current of 24 mA. The central portion (5 mm × 5 mm) of the powder bed was then hatched using a snake scan under the conditions listed in Table 3.

## 3. Results

### 3.1. Morphology of Ball-Milled Powders

Figure 1d–f show SEM images of the Inconel 718 PA powder after ball milling in air at a rotation speed of 800 min^−1^ for 10, 30, or 60 min. After ball milling for 10 min (Figure 1d), the shape of the PA powder particles was more distorted than that of the virgin PA powder particles (Figure 1a), but the particle size and particle size distribution were largely unchanged. However, after ball milling for 30 min (Figure 1e), the small and large powder particles had agglomerated, in addition to the distortion of the particle shape. Furthermore, after ball milling for 60 min (Figure 1f), the particle size had increased by several times compared to the size of the virgin powder particles due to agglomeration.

The SEM image in Figure 4 shows the surface morphology of a single particle of the ball-milled (30 min) powder in more detail. Compared to the virgin powder particles shown in Figure 1a–c, the surface of the ball-milled powder particles was deformed, indicating that the powder particles were plastically deformed during ball milling.

Figure 5 and Table 4 show the results of the particle size distribution analysis of the virgin and ball-milled (30 min) powders. Whereas the D90 value increased after ball milling, D10, D50, and the average particle size decreased after ball milling. In addition, the particle size distribution was broader after ball milling. These results indicate that the powder particles were crushed during ball milling and further reaggregated to increase the particle size, resulting in a broader particle size distribution.

### 3.2. Surface Chemical Composition of Powders

To investigate the effect of the ball milling treatment on the surface oxide film, XPS analysis was performed before and after ball milling to elucidate changes in the powder surface composition. Figure 6 shows the XPS profiles of the powders before and after ball milling for 30 min. The XPS profiles of the virgin and ball-milled powders contained peaks that corresponded to the major alloying elements (Ni 2p, Ti 2p, Al 2p, Cr 2p, Mo 3d, Nb 3d), in addition to carbon (C 1s) and strong oxygen (O 1s) peaks. The XPS profiles of the two powders are almost identical, with no differences in the peak positions, indicating that the chemical compositions and types of compounds on the surface of the powders were consistent, with almost no contamination of the ball and container components. Carbon was most likely present from the carbon tape used for sample preparation [15]. 

Figure 7 shows high-magnification Cr 2p, Ti 2p, Mo 3d, Nb 3d, Al 2p, and O 1s XPS profiles of the virgin Inconel 718 alloy powder. Cr, Ti, Mo, Nb, and Al are major oxide-forming elements. Notably, Cr, Ti, and Nb were detected in their oxide states (i.e., Cr_2_O_3_, TiO_2_, and Nb_2_O_5_, respectively), which reveals the types of oxides present on the surface of the powder. 

Ar gas ion etching was performed on the powder surface, and continuous XPS measurements were performed to obtain a chemical profile in the depth direction. We used SiO_2_ as the reference material for the etching rate and converted the etching rate to depth. Figure 8 shows the change in the atomic fraction of each element in the virgin PA powder depending on the etching depth. The oxygen concentration on the outermost surface (i.e., at an etching depth is zero) was as high as 40 at.%. As the depth increased slightly, the oxygen concentration exceeded 50 at.%. This proves that the powder surface is covered with an oxide film.

Focusing on the outermost surface, it is considered that the outer 2–3 nm (i.e., the region where the oxygen profile shows a peak) is the oxide film. The main component of the surface oxide film is likely to be Cr_2_O_3_, because of the sharp increase in Cr concentration between 0 and 1 nm, reaching a concentration of approximately 12 at.%. The Cr concentration then gradually increased as the depth increased further, reaching a concentration of approximately 21 at.% (18.8 wt.%) at a depth of 20 nm. The concentration of Ni, which is the main element of Inconel 718, also increased rapidly with depth; however, when Ni and Cr coexist, Cr_2_O_3_ is thermodynamically favored to form over Ni oxides [11]. In addition, from the tendencies of the Ti, Nb, and Al concentration profiles, with the concentrations increasing sharply near the surface and then decreasing in the depth direction, TiO_2_, Nb_2_O_5_, and Al_2_O_3_ are also likely to exist in the surface oxide film, although the low concentrations indicate that they exist in smaller amounts than Cr_2_O_3_. 

The O 1s peaks of the oxide films on the virgin and ball-milled (30 min) powders were deconvoluted into H_2_O, OH^−^, and O^2−^ bonding states, as shown in Figure 9 [13]. Before ball milling (Figure 9a), mainly OH^−^ and O^2−^ bonding states were observed. For metal oxides, the O^2−^ component is usually assigned to the oxide [16] (e.g., Cr–O–Cr bonds), while the OH^−^ component typically corresponds to the presence of hydroxides or adsorbed oxygen species [17], or to higher valency oxides [17]. In this case, the OH^−^ component likely arises from adsorbed water. Therefore, the intensity of the OH^−^ peak is proportional to the amount of adsorbed water, which, in turn, is expected to increase with the porosity of the surface oxide film. From Figure 9a,b, it can be seen that the intensity ratio of the OH^−^ peak to the O^2−^ peak decreased after ball milling. This suggests that the oxide film became denser, rather than being removed by ball milling. This may be because the ball milling was conducted in air rather than an inert gas such as argon.

### 3.3. Electrical Properties of Powders

#### 3.3.1. DC Electrical Resistivity of Virgin and Ball-Milled Powders

Figure 10 shows how the DC electrical resistivity of the virgin and ball-milled (10, 30, and 60 min) PA powders of Inconel 718 alloy changed during heating and cooling between room temperature and 800 °C. At room temperature, the electrical resistivity of the virgin powder was high, and comparable to those of insulating or dielectric materials. This high electrical resistivity is attributed to indirect contact between neighboring metal powder particles through the surface oxide films. Upon heating, the electrical resistivity of the virgin powder decreased, converging to the order of 10^−4^ Ω∙m, which is indicative of a metallic conductor. The electrical resistivity of oxides decreases upon heating, while that of metallic materials increases. Thus, the decreasing electrical resistivity of the virgin powder predominantly stems from the properties of the surface oxide film, rather than the internal properties of the metal particles. However, when cooling from 900 °C to room temperature after holding at 800 °C for 1 h, the resistivity of the virgin powder increased only slightly, and remained on the order of 10^−4^ Ω∙m even after returning to room temperature, which is eight orders of magnitude lower than that of the unheated virgin powder. Hence, there was a large hysteresis between the heating and cooling curves. This indicates that the thermal stability of the surface oxide film is extremely weak. The surface oxide film may become unstable due to heating, causing its insulating characteristics to change to a metallic nature. 

Meanwhile, the room temperature DC electrical resistivity of the ball-milled (10 min) powder was six orders of magnitude lower than that of the virgin powder before heating. This significant reduction in electrical resistivity may result from the ball milling process. For the 30- and 60-min ball-milled powders, the room temperature resistivity dropped by seven orders of magnitude to the order of 10^−3^ Ω∙m. This decrease in resistivity is reminiscent of the metal–insulator transition of Mott insulators [13]. From these experimental results, it can be concluded that ball milling can increase the electrical conductivity of the surface oxide film. A detailed discussion of these results is provided in Section 4.2. 

#### 3.3.2. AC Impedance of Virgin and Ball-Milled Powders

Figure 11a shows Nyquist diagrams of the virgin Inconel 718 PA powder measured by AC impedance spectroscopy [17,18,19] at room temperature and at 50, 100, 200, 300, 400, and 800 °C. The upward direction (↑) is the capacitive reactance component, and the absolute value of capacitance increases in this direction. The resistance value increases as the horizontal axis moves in the right direction (→). As shown in Figure 11a, the Nyquist plots measured between room temperature and 200 °C are semicircular. This shows that the impedance can be represented by a parallel circuit of the capacitor and resistance components. The measured values were much smaller at temperatures of ≥300 °C; therefore, an enlarged view is shown in Figure 11b. Furthermore, the Nyquist plots do not form semicircles; instead, they have a curved shape with a convex curvature in the direction of the horizontal axis. This demonstrates that the equivalent circuit at 300 °C consists of a series of capacitors and resistors. In the equivalent circuit, the value of the capacitor component decreases as the temperature increases.

Nyquist diagrams of the ball-milled (10, 30, and 60 min) powder are shown in Figure 12. For all the ball-milled powders, Nyquist curves appear below the origin of the vertical axis. This indicates that the capacitive reactance component disappeared and changed to an inductive reactance component. Furthermore, for the powder subjected to ball milling for 10 min (Figure 12a), the Nyquist curve moves toward the origin in the horizontal direction (←) as the temperature rises from room temperature to 800 °C. This indicates that the resistance component becomes smaller as the temperature increases. At ball milling times of 30 and 60 min (Figure 12b,c), the shape of the Nyquist curve is almost the same as that for powders with a ball milling time of 10 min. However, the position of the Nyquist curve hardly changes with the measurement temperature. That is, the temperature dependence of the resistance component disappears as the ball milling time increases. 

### 3.4. Smoke Detection Tests

From the SEM images of the Inconel 718 alloy PA powder shown in Figure 1 and Figure 4, the ball milling treatment subjects the powder surface to plastic deformation and distortion. If the ball milling time is too long, the powder particles aggregate and coarsen (Figure 1f). As shown in Figure 4a,b, while the surface of the powder was distorted after ball milling for 30 min, the powder particles exhibited negligible agglomeration and their shape remained almost spherical. Moreover, although the data are not shown here, the results of flowability tests were almost the same as those of the virgin powder. Therefore, the powder with a ball milling time of 30 min is likely to be suitable for use in actual PBF-EBM processes. Consequently, the 30-min ball-milled powder was selected for use in smoke detection tests.

The smoke detection tests were performed at an irradiation current of *I* = 20 mA, beam irradiation frequency of *f*_1_ = 100 Hz, and beam dwell time of 1/*f*_2_ = 0.1 ms (*f*_2_ = 10 kHz). Table 5 summarizes the typical results for the virgin and ball-milled (30 min) powders, measured at a beam diameter (full width at half maximum; FWHM) of 1.8 mm, and preheating temperatures between room temperature (23 °C) and 800 °C (chosen arbitrarily). The results of the smoke detection test are shown in the rightmost column of Table 5. N.D. indicates that smoking was not detected, while the values indicate the irradiation time at which smoking began. Under the abovementioned test conditions, smoking of the virgin powder occurred at all temperatures between room temperature and 700 °C, and a preheating temperature of 800 °C was required to suppress smoking. On the other hand, the ball-milled powder did not smoke even at room temperature under the same test conditions. Thus, the ball milling treatment significantly reduced the preheating temperature required to suppress smoking. Moreover, smoking did not occur even without preheating. 

Figure 13 shows a series of snapshots of a smoke test performed on virgin powder without preheating. The beam irradiation pulse frequency and dwell time were 100 Hz and 0.1 ms, respectively. A dent was formed in the powder layer about 72.8 ms after irradiation was initiated (Figure 13b), and the powder was clearly scattered after a further 17 ms (at 89.6 ms; Figure 13c). The powders became completely scattered 30 ms after smoking was first observed. Smoking lasted for 310 ms, after which the powder was completely removed from the groove on the stainless steel plate (Figure 13h).

Figure 14 shows a series of snapshots of a smoke test performed on virgin powder with a base plate preheating temperature of 600 °C. A glowing red melt pool was formed on the powder bed 200 ms after irradiation was initiated (Figure 14b). After that, the melt pool gradually expanded; however, the powder around the melt pool began to move at 497.2 ms (Figure 14c), and smoke was completely generated at 510 ms (Figure 14d). At 600 ms, the melt pool was blown away with the smoke (Figure 14f). At 800 ms, the powder in the groove completely dispersed (Figure 14g), and at 900 ms, the metallic luster of the groove in the base plate could be clearly observed (Figure 14h).

Figure 15 shows a series of snapshots of a smoke test performed on virgin powder with a base plate preheating temperature of 800 °C. The irradiated powder began to glow red approximately 100 ms after irradiation was initiated (Figure 15c), with a melt pool forming at 150 ms (Figure 15d). After that, the size of the melt pool gradually increased. Notably, smoking did not occur even when the test end time (1400 ms) was reached. Therefore, when the virgin powder is irradiated under the current beam conditions, it seems that smoking does not occur when the base plate is preheated to 800 °C. It is estimated that the temperature at which smoking of the virgin powder occurs is approximately 700 °C. 

Figure 16 shows a series of snapshots of a smoke test performed on ball-milled (30 min) powder without preheating. Similar to the smoke test of the virgin powder with preheating at 800 °C (Figure 15), the powder bed began to turn red approximately 100 ms after irradiation was initiated (Figure 16c), with a melt pool forming at 150 ms (Figure 16d). The melt pool continued to expand until the end of the test without generating smoke. This shows that smoking can be avoided by using ball-milled powders, even without preheating.

### 3.5. Hatching Test of Ball-Milled (30 Min) Powder

In Section 3.4, it was shown that smoking does not occur even if an unheated bed of ball-milled powder is subjected to electron beam irradiation. This result suggests that ball milling of raw powders is an effective method of lowering the smoke generation temperature. To verify whether a bed of ball-milled powder could be melted by electron beam irradiation without smoking or spattering after preheating the base plate to 500 °C, hatching (powder bed melting) tests were conducted using the 30-min ball-milled powder. This powder was selected because its particle shape and size distribution are almost the same as those of the untreated powder. 

The experimental conditions of the hatching test are shown in Table 3. The beam current, beam scanning speed, line offset, and beam diameter were 1.2 mA, 200 mms^−1^, 0.2 mm, and 350 μm, respectively. Figure 17 shows a series of snapshots of the hatching test. No smoking was observed, and hatching was performed without sparking or spattering. Considering that a preheating temperature of approximately 1050 °C is typically used in PBF-EBM processes with ordinary Inconel 718 alloy powder [20,21], it is noteworthy that the powder bed can be melted by a hatching beam at a preheating temperature of 500 °C. When the powder bed began to melt (Figure 16b), multiple red dots and lines appeared around the powder specimen. These artifacts are reflections of the light from the red-hot melted region on the observation window glass of the door of the EBM machine, not spattering or sparking. This confirms that the ball-milled (30 min) powder is suitable for use in PBF-EBM processes.

The above results demonstrate that it is possible to reduce the smoking temperature of Inconel 718 alloy PA powder to room temperature by ball milling for 30 min. Furthermore, powder bed melting can be achieved at a preheating temperature of 500 °C by a normal hatching process, without causing spattering and sparking that can lead to defects in the built parts. Considering that the PBF-EBM process of Inconel 718 alloy powder usually requires a preheating temperature of 1000 to 1050 °C [20,21] or higher to avoid smoking, the “ball milling effect” described in this study is highly valuable for developing techniques to suppress smoking in PBF-EBM without relying on preheating.

## 4. Discussion

### 4.1. Origin of Temperature Dependence of Electrical Resistivity of Inconel 718 Alloy Powder

The XPS analysis in Section 3.2 demonstrated that the surface oxide film of the PA powder of Inconel 718 alloy is composed of Cr_2_O_3_, TiO_2_, and Nb_2_O_5_. However, the main component of the surface oxide film is Cr_2_O_3_ [22]. To investigate the thermal stability of Cr_2_O_3_, the temperature dependence of the resistivity of Cr_2_O_3_ powder was investigated using the method described in Section 2.4 (Figure 10). The Cr_2_O_3_ powder used for the measurement was a Cr_2_O_3_ reagent (micron-scale particles) manufactured by Wako Pure Chemical Industries. Figure 18 shows the temperature dependence of the electrical resistivity of the Cr_2_O_3_ powder between room temperature and 800 °C. At room temperature, the measured electrical resistivity value exceeded 10^7^ Ω∙m, which is the measurement limit of the equipment used; therefore, the actual electrical resistivity value could not be obtained. Nevertheless, this result demonstrates that the Cr_2_O_3_ powder is a dielectric material. The resistivity remained above 10^7^ Ω∙m until the temperature exceeded 300 °C. Then, the resistivity decreased monotonically with the subsequent temperature rise, reaching the order of 10^1^ Ω∙m at 800 °C. As the temperature decreased again from 800 °C, the resistivity rose reversibly. When the temperature dropped to 200 °C, the resistivity value again exceeded 10^7^ Ω∙m; thus, the Cr_2_O_3_ powder returned to its initial dielectric state. This suggests that the thermal stability of Cr_2_O_3_ is high, even when held at a temperature of 800 °C for 1 h [23]. 

The room temperature resistivity of the virgin Inconel 718 alloy powder was on the order of 10^4^ Ω∙m, as shown in Figure 10. Thus, it can be classified as a semiconductor. On the other hand, after heating to 800 °C and cooling to room temperature, the electrical resistivity was on the order of 10^−4^ Ω∙m, which is characteristic of a metallic structure. This suggests that the electrical properties of the oxide film on the powder surface change irreversibly from dielectric to metallic after heating to a high temperature. From this, it can be inferred that the Cr_2_O_3_ oxide film on the powder surface had poor thermal stability. The difference in thermal stability between the Cr_2_O_3_ powder and Cr_2_O_3_ surface oxide film is likely to be caused by the difference in scale (micron-sized particles vs. nanometer-scale film). The surface oxide film on the Inconel 718 alloy powder was several nanometers thick. As the temperature increases, oxygen atoms move from the oxide film into the bulk of the powder particle by thermal diffusion. As a result, the Cr_2_O_3_ in the oxide film becomes Cr_2–*x*_O_3–*x*_, which is oxygen-deficient. At the same time, its electrical properties change from dielectric to metallic. This thermal diffusion of oxygen is irreversible, which is why the thermal stability of the surface oxide film on the Inconel 718 alloy powder is poor.

### 4.2. Electrical Properties of Oxide Film and Effect of Ball Milling

Figure 10 shows the temperature dependence of the electrical resistivity of the virgin and ball-milled (10, 30, and 60 min) Inconel 718 PA powders. The room temperature electrical resistivity of the 10-min ball-milled powder was on the order of 10^−3^ Ω·m. Consequently, ball milling for 10 min causes the resistivity to decrease by seven orders of magnitude from 10^4^ to 10^−3^ Ω·m. As mentioned in the Section 2.2, the ball milling process was conducted in air; therefore, even if the surface oxide film is peeled off during ball milling to expose the fresh metal surface, the surface of the alloy powder will immediately oxidize again. Therefore, the electrical properties of the surface oxide film of the Inconel 718 PA powder are changed from dielectric (semiconductor) to metallic by the ball milling treatment. The XPS analysis of the surface oxide film (see Figure 9 and Section 3.2) demonstrated that the intensity of the O^2−^ peak increased after ball milling, which indicates that the oxide film becomes denser. Thus, the XPS analysis also supports the idea that the oxide film on the surface of the alloy powder is electrically transferred from dielectric to metal by ball milling.

In this section, we consider how ball milling affected the electrical characteristics of the surface oxide film on the Inconel 718 alloy PA powder. First, let us consider the resistance component of the alloy powder bulk and the resistance, capacitance, and inductance components of the surface oxide film, as obtained by AC impedance measurements [17,18,19] before ball milling. Figure 11a shows the Nyquist plots of the virgin powder obtained at room temperature and at 50, 100, and 200 °C. Fitting this Nyquist diagram to the impedance equation of the dielectric relaxation type (Cole–Cole relaxation) shown in Equation (1), the temperature dependence of the resistance and capacitor components of the equivalent circuit can be obtained.
(1)$$Z=R_{\mathrm{metal}}+\frac{R_{\mathrm{oxide}}}{{\left(1+\left(i\omega R_{\mathrm{oxide}}C_{\mathrm{oxide}}\right)\right)}^{p}}=Z^{\prime }-iZ^{\prime \prime },$$
where *R*_metal_ and *R*_oxide_ are the resistance components of the alloy powder bulk and surface oxide film, respectively; *C*_oxide_ is a capacitance component of the surface oxide film; *i* is an imaginary unit; *ω* is the AC frequency; and *p* is an exponent between 0 and 1. The time constant (relaxation time; *τ*) can be obtained from the product of *R*_oxide_ and *C*_oxide_ (*τ* = *R*_oxide_ × *C*_oxide_). Figure 19 shows the results of fitting Equation (1) to the Nyquist diagrams shown in Figure 11a. The measured Nyquist diagrams fit well with Equation (1). Therefore, it can be determined that the equivalent circuit of the powder layer of the Inconel 718 alloy (Figure 20) forms a parallel circuit with the resistance (*R*_oxide_) and capacitance (C_oxide_) of the surface oxide film coupled in series with the electric resistance of the alloy powder bulk (*R*_metal_).

Table 6 summarizes the resistance (*R*), capacitance (*C*), and relaxation times (*τ*) obtained from the fitting of Equation (1). The virgin Inconel 718 alloy powder has a capacitance component, *C*_oxide_, borne by the surface oxide film. This capacitance component leads to smoking, because negative charges accumulate in the alloy powder particles during electron beam irradiation because of the capacitance of the surface oxide film.

The AC impedance measurements of the ball-milled (30 min) Inconel 718 alloy powder had a different outcome. The resistance component of the bulk alloy powder (*R*_metal_) and the resistance, capacitance, and inductance components of the surface oxide film (*R*_oxide_, *C*_oxide_, and *L*_oxide_, respectively) were obtained. Figure 21a–g show Nyquist plots of the 30-min ball milled powder at room temperature and at 50, 100, 200, 300, 400, and 800 °C, respectively. A fitted curve is shown based on the equivalent circuit shown in Figure 22. The experimental and fitted curves were very similar, showing that the fitting was successful. This indicates that the AC impedance of the 30-min ball-milled powder can be represented by the equivalent circuit shown in Figure 22. That is, the equivalent circuit of the 30-min ball-milled powder can be represented by a parallel circuit of *R*_oxide_ and *L*_oxide_ with *R*_metal_ coupled in series. Hence, compared with the equivalent circuit of the Inconel 718 alloy powder before ball milling (Figure 20), *C*_oxide_ is replaced with *L*_oxide_ in the equivalent circuit after ball milling.

The values of *R*_metal_, *R*_oxide_, and *L*_oxide_ after ball milling are summarized in Table 7. The data indicate that the capacitance component disappears after ball milling and is replaced with the inductance component. At present, it is unclear what kind of change in physical properties causes the inductive reactance of the surface oxide film, *L*_oxide_. Therefore, further research is required. Furthermore, the value of *R*_oxide_ reduced the order of 10^5^ Ω to 19.2 Ω after ball milling. This indicates that the electrical properties of the surface oxide film change from dielectric to metallic properties during ball milling. This agrees with the conclusion obtained from the DC electrical resistivity results described in Section 3.3.1 regarding the dielectric–metal transition that occurs during ball milling. 

Overall, the AC impedance measurement results confirm that the surface oxide film of the virgin powder behaves like a dielectric material and functions as a capacitor. When the powder bed is irradiated with an electron beam, negative charges accumulate within the powder particles because the surface oxide film acts as a capacitor, leading to smoking. However, when subjected to mechanical stimulation by ball milling, the electrical properties of the surface oxide film shift from dielectric to metallic. As a result, the oxide film loses its capacitor function, which means that smoking does not occur upon electron beam irradiation.

### 4.3. Suppression of Powder Bed Smoking

The phenomenon of smoking, in which the powder bed is scattered by electron beam irradiation, is unique to EBM; that is, it is not found in laser beam manufacturing processes. Controlling this smoking behavior is expected to drastically expand the process window of EBM technology. To develop equipment and materials technologies that can control smoke generation during EBM, it is necessary to understand the mechanism of smoke generation in detail. In this section, we consider the mechanisms of charging and discharging the powder bed during electron beam irradiation and consider guidelines for optimizing the materials technology, including the process conditions and powder properties of EBM, to suppress smoking.

#### 4.3.1. Construction of Equivalent Circuit of EBM System

Figure 23 shows the equivalent circuit of the EBM basic system (electron beam/powder bed/base plate) by applying the equivalent circuit of the alloy powder obtained from the electrical resistance and impedance measurements in Section 4.2 to the powder bed. In the equivalent circuit shown in Figure 23b, if the electrical resistivities of the surface oxide film (*R*_oxide_), bulk metal (*R*_metal_), and base plate (*R*_BP_) are approximately the same (*R*_oxide_ ≈ *R*_metal_ ≈ *R*_BP_), the powder bed and base plate can be heated without smoking, even if the powder bed is subjected to electron beam irradiation, as shown in Figure 23a. However, if *R*_oxide_ ≫ *R*_metal_, negative charges will accumulate in the powder bed during electron beam irradiation, leading to smoke generation. In fact, as shown in Table 2, the condition of *R*_oxide_ ≫ *R*_metal_ is satisfied in the alloy powder used for practical EBM, which means that smoking is unavoidable in practical alloy powders. Therefore, the materials technology required to avoid smoking even when the alloy powder is irradiated with an electron beam is to eliminate *C*_oxide_ or reduce *R*_oxide_ to a value comparable to that of *R*_metal_.

#### 4.3.2. Powder Bed Charge Accumulation and Smoking Conditions

Cordero et al. [24] determined the maximum charge generated in powder particles of a powder bed during preheating from charge *Q*(*t*) by solving the initial value problem, and found that smoking occurs when the following equation holds:(2)$$\rho g\frac{4}{3}\pi R^{3}\leq \frac{\pi R^{2}}{16\varepsilon_{0}}{\left[J\eta \tau \left(\frac{1-e^{-\frac{kD}{v\tau }}}{1-e^{-\frac{T}{\tau }}}\right)\right]}^{2}f\left(\frac{h}{R}\right),$$
where *g* is the gravitational acceleration, *ρ* is the theoretical density of the powder, *R* is the radius of the powder, *ε*_0_ is the dielectric constant of a vacuum, *J* is the current density of the electron beam, *η* is the energy absorption efficiency, *τ* is the discharge time of the accumulated charge (time constant), *D* is the electron beam diameter, and *T* is the time (cycle) until the electron beam returns to its initial position. *f*(*h*/*R*) is a geometric factor, wherein *h* is the thickness of the surface oxide film on a powder particle with radius *R*. In Equation (2), the left side is the weight of the powder particles, while the right side is the electrostatic force (Coulomb force); therefore, the condition for smoke generation is that the electrostatic force acting on the powder must be greater than or equal to the weight of the powder. 

Cordero et al. [24] substituted the actual beam condition that enables building under the PBF-EBM condition of Ti–6Al–4V alloy powder into Equation (2) to obtain the value of *τ* as a building condition that does not generate smoke. Their results indicated that the value of *τ* should be approximately 0.5 ns or less. As summarized in Table 5, for Inconel 718 alloy powder, the values of *τ* at room temperature and at 50, 100, and 200 °C were 2.84, 2.45, 2.12, and 0.918 μs, respectively. The value of *τ* decreases as the temperature increases, but it is higher than 0.5 ns, which was previously considered the limit for avoiding smoking. As mentioned in Section 3.3.1, the electrical properties of the alloy powders decline sharply with an increase in temperature. Inconel 718 alloy powder exhibits metallic characteristics at temperatures above 700 °C, with low electrical resistivity. In addition, *C*_oxide_ tends toward zero as the temperature rises, as demonstrated by the impedance measurement results (Figure 11b). This means that the capacitance of the surface oxide film of the alloy powder disappears. In this case, *τ* = *R*_oxide_ × *C*_oxide_ = 0, and from Equation (2), the electrostatic force does not act on the powder, and smoking does not occur. In the EBM technology, the reason for preheating to suppress the generation of smoke is to eliminate the capacitive component of the surface oxide film of the alloy powder and reduce the value of *τ* to zero. Therefore, if the individual alloy powder particles have fully metallic characteristics—including the surface oxide film—and if all the individual powder particles are in contact with each other and electrically conductive to the base plate, the smoking phenomenon will be avoided. It is not necessary to sinter the powder in the powder bed by the residual heat generated by heating the base plate by electron beam irradiation, but it is important that the electrical contact between the metal powders and the capacitor component of the surface oxide film disappears to make *τ* zero.

### 4.4. Metal–Insulator Transition by Ball Milling of Inconel 718 Alloy Powder 

The electronic conductivity of oxides of transition metals with partially filled 3d and 4d orbitals is very low, and their properties cannot be explained by the free electron model [25]. To understand these compounds, N.F. Mott succeeded in explaining how electronic conductors can become electrical insulators by considering the electrostatic interaction between electrons [26,27]. These materials are known in the field of solid-state physics as Mott insulators. In the Mott insulator model, most of the electrons are localized near their parent atom, and when moving electrons occupy the atom, there is a significant decrease in conductivity. This can be regarded as a competition between the kinetic energy of electrons and the Coulomb repulsive force. If Coulomb repulsion is large, the material becomes an insulator. Therefore, a metal–insulator transition can occur by changing the kinetic energy of the electrons or the Coulomb repulsive force. Many 3d and 4d transition metal oxides, such as TiO_2_, V_2_O_3_, and Cu_2_O, exhibit this type of metal–insulator transition [28,29,30]. Such Mott insulators appear not only in bulk materials but also in thin films. A recent study of V_2_O_3_ thin films at room temperature showed that the metal–insulator transition occurred when temperature, element doping, and pressure were applied. The metal–insulator transition in V_2_O_3_ is caused by heteroepitaxy-induced continuous lattice distortion. The in-plane controlled epitaxial strain of the V_2_O_3_ thin film stabilizes not only the structure of the thin film, but also its electrical and optical properties in various intermediate states between the metallic and insulating phases that it is not possible to achieve with bulk materials. This causes a significant change in room temperature resistivity (Δ*R*/*R* up to 100,000%). 

According to the XPS surface analyses of the powders, as discussed in Section 3.2, the oxide film formed on the surface of the Inconel 718 alloy powder in this study is mainly composed of Cr_2_O_3_ oxide. To date, no studies have been conducted on Cr_2_O_3_ oxides; however, it is possible that electrical properties, like those of V_2_O_3_, will appear in Cr_2_O_3_ films. Furthermore, it is highly possible that the metal–insulator transition occurs in Cr_2_O_3_ due to the introduction of strain by ball milling. Therefore, by ball milling the Inconel 718 alloy powder, the resistivity was reduced by seven orders of magnitude, and the capacitance component of the oxide film disappeared. This is because the electrical conduction of the Cr_2_O_3_ oxide film, which is a Mott insulator, undergoes a metallic transition due to the lattice strain introduced by the ball milling treatment.

## 5. Conclusions

Smoking, which is a problem in electron beam additive manufacturing, is generated by the Coulomb repulsive force acting between the powder particles, because they become negatively charged under electron beam irradiation. In this study, using Inconel 718 alloy PA powder, we investigated how preheating suppressed smoking by changing the electrical properties of the powder bed. Furthermore, it was predicted that the natural oxide film formed on the powder surface would undergo a dielectric–metallic transition by mechanical stimulation such as ball milling. We investigated whether mechanical stimulation of the alloy powder could be used to suppress smoking without preheating. The results are summarized below.

Inconel 718 alloy PA powder that was ball milled for 10 min in air has a more distorted powder shape than the virgin PA powder, but the particle size and particle size distribution were mostly unchanged. However, after 30 min of ball milling, agglomeration of the small and large powder particles was observed, in addition to deformation of the particle shape. Furthermore, after 60 min of ball milling, the powder particle size increased to several times the original particle size due to the agglomeration of the powder.From XPS analysis and depth profiles of the virgin Inconel 718 alloy PA powder, it was found that the Cr_2_O_3_, TiO_2_, and Nb_2_O_5_ exist in the oxide film on the surface of the powder; however, the film mainly comprises Cr_2_O_3_. This proves that the powder surface is covered with an oxide film. In addition, it was considered that TiO_2_, Nb_2_O_5_, and Al_2_O_3_ were also present in the oxide film in small amounts. After ball milling for 30 min, the O^2−^ peak became relatively higher than the OH^−^ peak, suggesting that the density of the oxide layer increased after ball milling for 30 min.The electrical resistivity of the Inconel 718 alloy PA powder was on the order of 10^4^ Ω∙m at room temperature and decreased with increasing temperature. This indicates that the origin of the electrical resistivity of the powder stems from the surface oxide film, which is comparable to that of an insulator or dielectric material. In the process of cooling to room temperature after holding at 800 °C for 1 h, the resistivity of the virgin powder remained on the order of 10^−4^ Ω∙m even after returning to room temperature, resulting in a large hysteresis. This giant reduction in electrical resistivity—by eight orders of magnitude—indicates that the thermal stability of the oxide film on the surface of the Inconel 718 PA powder particles is extremely low. Therefore, it is suggested that the surface oxide film of the powder particles becomes unstable due to heating, whereby its electrical properties adopt a metallic nature.The Nyquist plots of the Inconel 718 alloy PA powder are depicted as semicircle Cole–Cole plots in the lower temperature region, wherein the oxide surface film has a capacitor and a resistor component. The equivalent circuit of the Inconel 718 alloy PA powder was determined to be a parallel circuit of the resistance (*R*_oxide_) and capacitance (*C*_oxide_) of the surface oxide film coupled in series with the electric resistance of the alloy powder bulk (*R*_metal_). The resistance (*R*_oxide_) and capacitance (*C*_oxide_) components of the surface oxide film disappeared at approximately 400 °C, which is associated with the collapse of the oxide film into the metal.Smoking is caused by the negative charge of the metal powder by electron beam irradiation, and is closely related to the dielectric relaxation behavior of the oxide film on the powder surface. The negative charge of the metal powder arises from the capacitive component of the oxide film on the surface of the metal powder. By eliminating the capacitive component of the surface oxide film by heating or mechanical stimulation, the dielectric relaxation time (time constant) tends toward zero, resulting in the suppression of the charge of the powder, thereby avoiding smoking.The capacitive component of the oxide film on the powder surface disappears due to ball milling of the oxide film, and at the same time, it transitions from an insulator to a metal, associated with the metal–insulator transition manifested in thin-film V_2_O_3_ (a Mott insulator). By mechanically stimulating the alloy powder (for example, by ball milling), it was demonstrated that smoking did not occur after ball milling, even if the powder bed was not preheated before being irradiated with an electron beam. From the above results, a PBF-EBM process without preheating is possible.

## 6. Patents

Technology Research Association for Future Additive Manufacturing (TRAFAM); Tohoku University. Powder for metal additive manufacturing, method for producing same, additive manufacturing device, and control program. Patent No. PCT/JP2019/010679, 14 March 2019.

## Figures and Tables

**Figure 1 materials-14-04662-f001:**
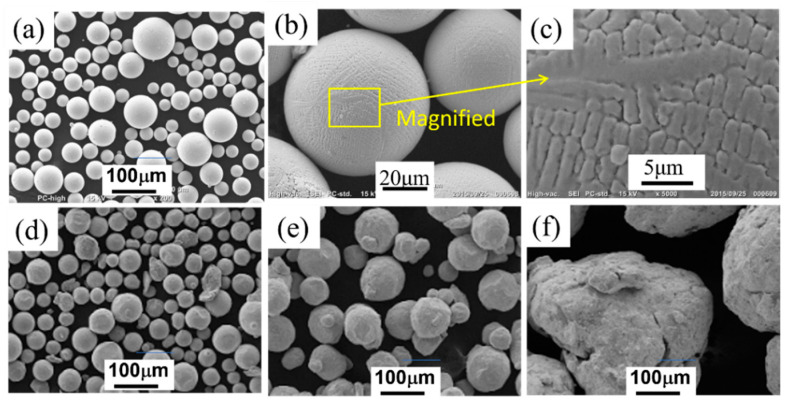
Scanning electron microscopy (SEM) images of Inconel 718 PA powder used in electrical resistivity and impedance measurements: (**a**) Virgin powder; (**b**) Magnified image of virgin powder; (**c**) Surface morphology of virgin powder; (**d**) Powder after ball milling for 10 min; (**e**) Powder after ball milling for 30 min; (**f**) Powder after ball milling for 60 min.

**Figure 2 materials-14-04662-f002:**
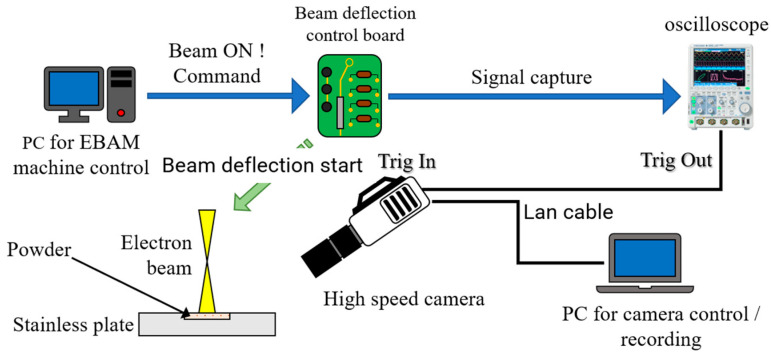
Schematic of the smoking experiment equipment.

**Figure 3 materials-14-04662-f003:**
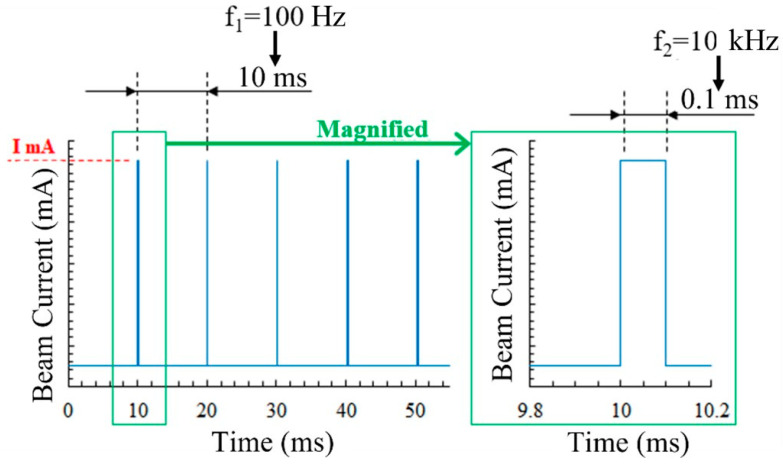
Beam irradiation pattern in smoke detection tests.

**Figure 4 materials-14-04662-f004:**
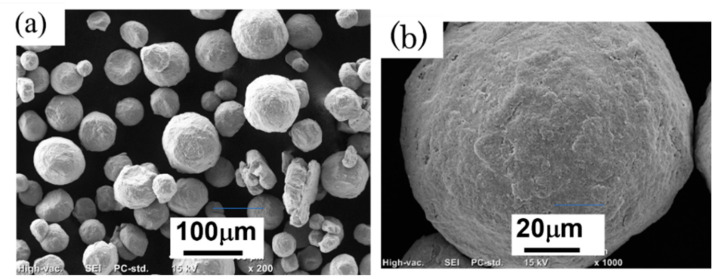
SEM images of Inconel 718 PA powder after ball milling for 30 min: (**a**) Morphology of powder particles; (**b**) Detailed surface morphology of one powder particle.

**Figure 5 materials-14-04662-f005:**
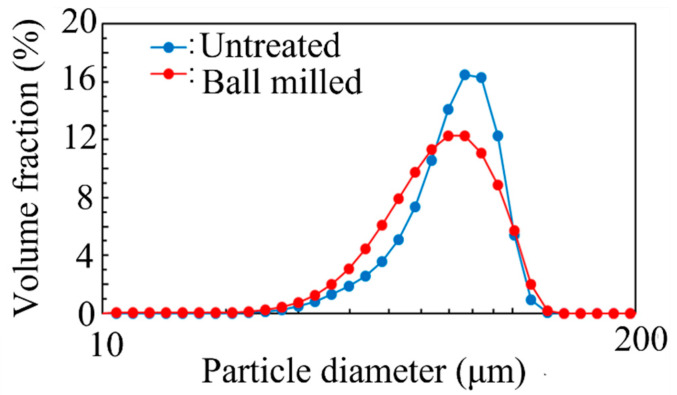
Particle size distribution of Inconel 718 PA powder before and after ball milling for 30 min.

**Figure 6 materials-14-04662-f006:**
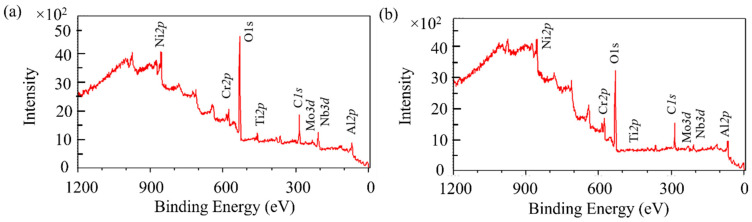
X-ray photoelectron spectroscopy (XPS) survey spectra of Inconel 718 PA powders: (**a**) Virgin powder; (**b**) Ball-milled (30 min) powder.

**Figure 7 materials-14-04662-f007:**
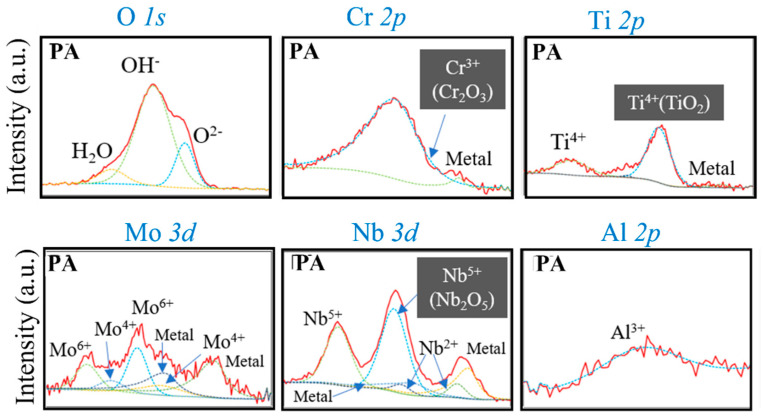
High-magnification XPS profiles of the Cr 2p, Ti 2p, Mo 3d, Nb 3d, Al 2p, and O 1s peaks of virgin Inconel 718 PA powder.

**Figure 8 materials-14-04662-f008:**
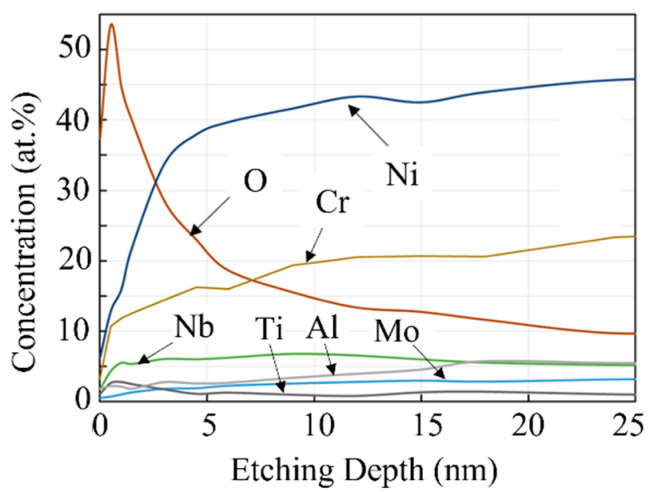
XPS depth profiles of virgin Inconel 718 PA powder.

**Figure 9 materials-14-04662-f009:**
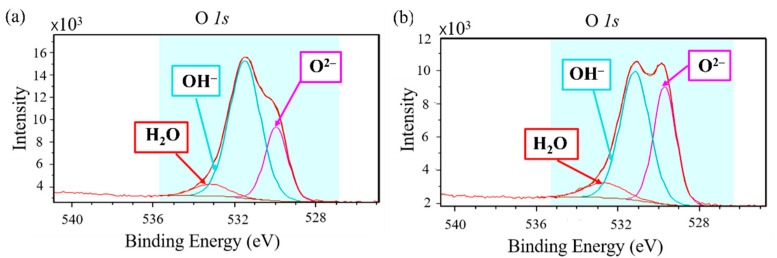
High-magnification narrow-scan O 1s profiles of Inconel 718 PA powder: (**a**) Virgin powder; (**b**) Ball-milled (30 min) powder.

**Figure 10 materials-14-04662-f010:**
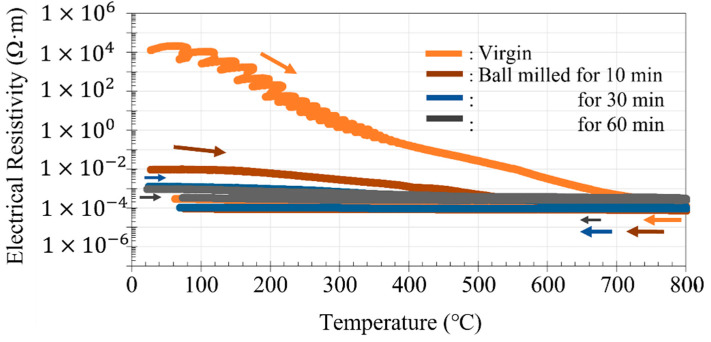
Change in direct current (DC) electrical resistivity of virgin and ball-milled (10, 30, and 60 min) Inconel 718 PA powder during heating (→) and cooling (←) between room temperature (23 °C) and 800 °C.

**Figure 11 materials-14-04662-f011:**
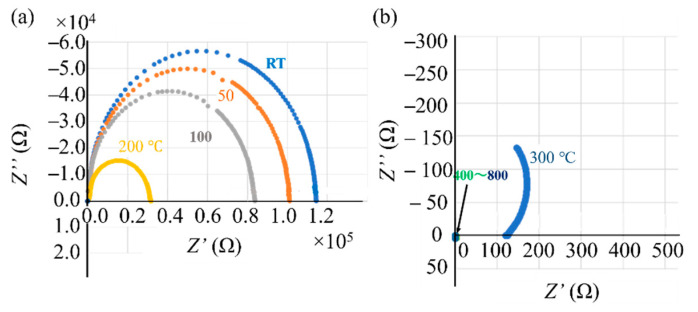
Nyquist diagrams of virgin Inconel 718 PA powder measured by alternating current (AC) impedance spectroscopy between room temperature (23 °C) and 800 °C: (**a**) Nyquist plots measured at room temperature and at 50, 100, and 200 °C; (**b**) Nyquist plots measured at 300, 400, and 800 °C (enlarged view).

**Figure 12 materials-14-04662-f012:**
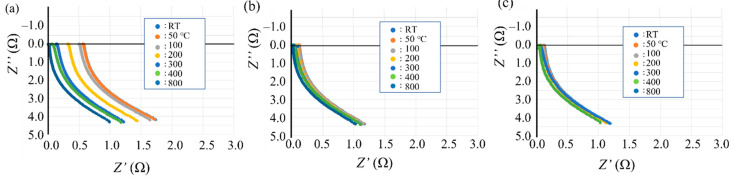
Nyquist diagrams of ball-milled Inconel 718 PA powder measured by AC impedance spectroscopy at room temperature (23 °C) and at 50, 100, 200, 300, 400, and 800 °C: (**a**) Ball milling time of 10 min; (**b**) Ball milling time of 30 min; (**c**) Ball milling time of 60 min.

**Figure 13 materials-14-04662-f013:**
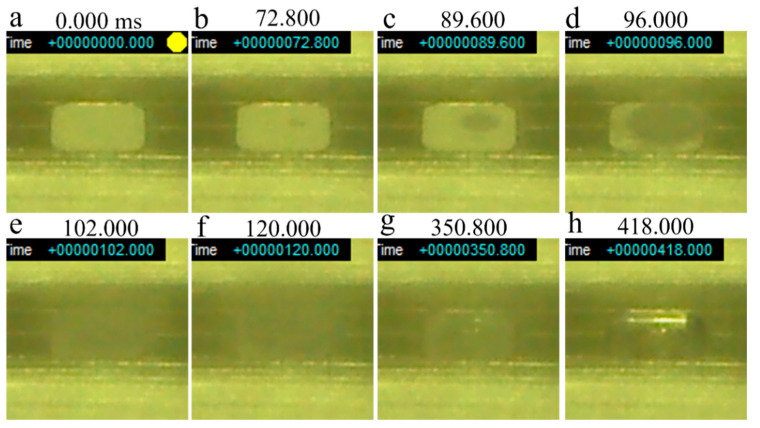
Series of snapshots of a smoke test performed on virgin Inconel 718 alloy PA powder without preheating: (**a**) 0 ms; (**b**) 72.8 ms; (**c**) 89.6 ms; (**d**) 96 ms; (**e**) 102 ms; (**f**) 120 ms; (**g**) 350.8 ms; (**h**) 418 ms. Beam irradiation pulse frequency (*f*_1_): 100 Hz; Beam dwell time (1/*f*_2_): 0.1 ms; Beam current (*I*): 20 mA; Beam diameter: 1.8 mm.

**Figure 14 materials-14-04662-f014:**
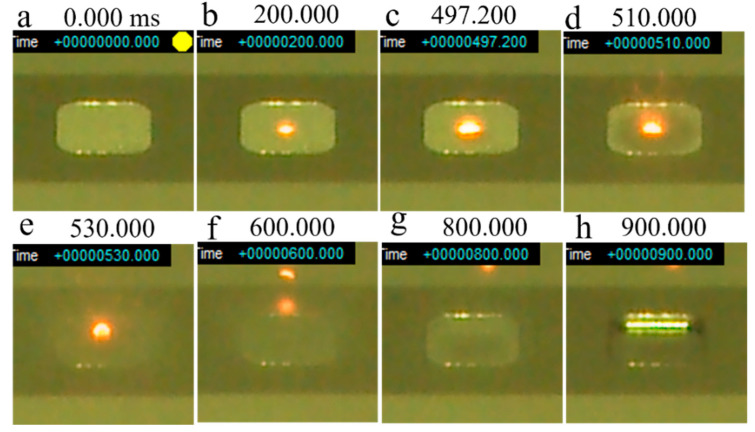
Series of snapshots of a smoke test performed on virgin Inconel 718 alloy PA powder with preheating at 600 °C: (**a**) 0 ms; (**b**) 200 ms; (**c**) 497.2 ms; (**d**) 510 ms; (**e**) 530 ms; (**f**) 600 ms; (**g**) 800 ms; (**h**) 900 ms. Beam irradiation pulse frequency (*f*_1_): 100 Hz; Beam dwell time (1/*f*_2_): 0.1 ms; Beam current (*I*): 20 mA; Beam diameter: 1.8 mm.

**Figure 15 materials-14-04662-f015:**
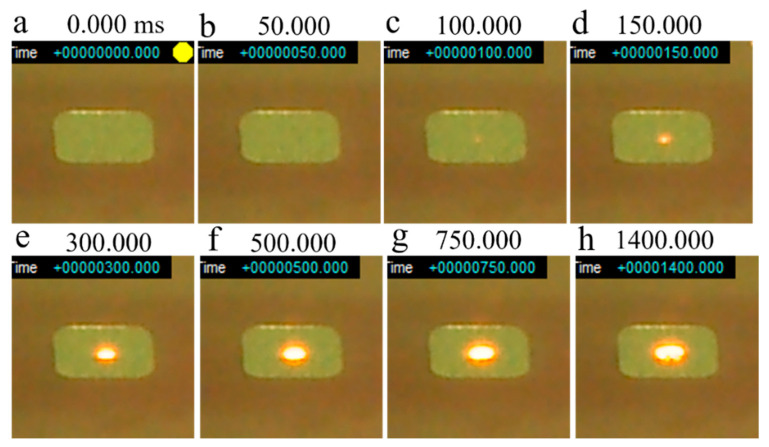
Series of snapshots of a smoke test performed on virgin Inconel 718 alloy PA powder with preheating at 800 °C: (**a**) 0 ms; (**b**) 50 ms; (**c**) 100 ms; (**d**) 150 ms; (**e**) 300 ms; (**f**) 500 ms; (**g**) 750 ms; (**h**) 1400 ms. Beam irradiation pulse frequency (*f*_1_): 100 Hz; Beam dwell time (1/*f*_2_): 0.1 ms; Beam current (*I*): 20 mA; Beam diameter: 1.8 mm.

**Figure 16 materials-14-04662-f016:**
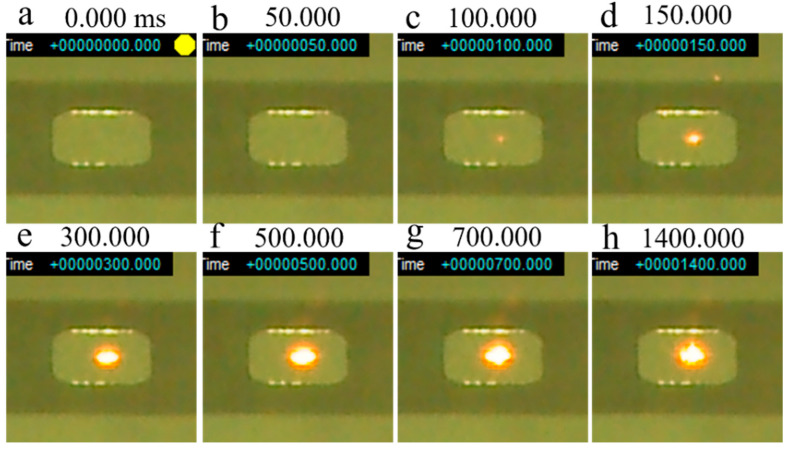
Series of snapshots of a smoke test performed on ball-milled (30 min) Inconel 718 alloy PA powder without preheating: (**a**) 0 ms; (**b**) 50 ms; (**c**) 100 ms; (**d**) 150 ms; (**e**) 300 ms; (**f**) 500 ms; (**g**) 700 ms; (**h**) 1400 ms. Beam irradiation pulse frequency (*f*_1_): 100 Hz; Beam dwell time (1/*f*_2_): 0.1 ms; Beam current (*I*): 20 mA; Beam diameter: 1.8 mm.

**Figure 17 materials-14-04662-f017:**
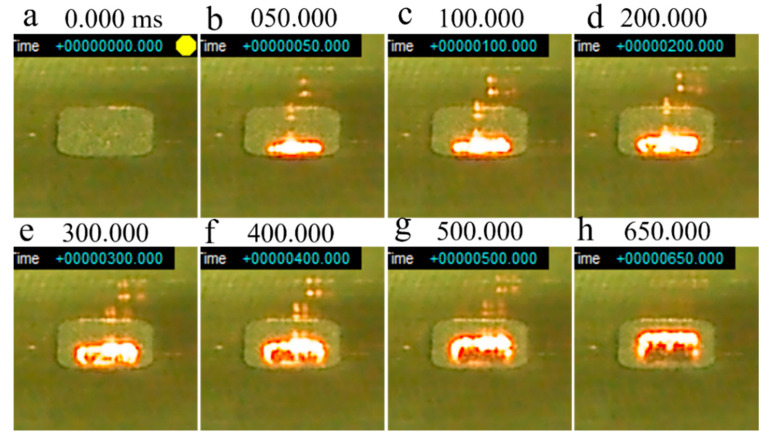
Series of snapshots of a hatching test performed on ball-milled (30 min) Inconel 718 alloy PA powder with preheating at 500 °C: (**a**) 0 ms; (**b**) 50 ms; (**c**) 100 ms; (**d**) 200 ms; (**e**) 300 ms; (**f**) 400 ms; (**g**) 500 ms; (**h**) 650 ms. Beam current: 1.2 mA; Beam scanning speed: 200 mm·s^−1^; Line offset: 0.2 mm; Beam diameter: 350 μm.

**Figure 18 materials-14-04662-f018:**
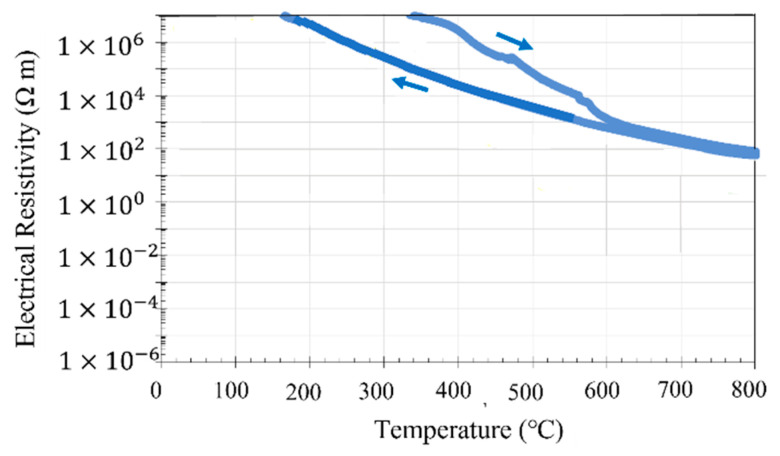
Temperature dependence of the electrical resistivity of Cr_2_O_3_ powder.

**Figure 19 materials-14-04662-f019:**
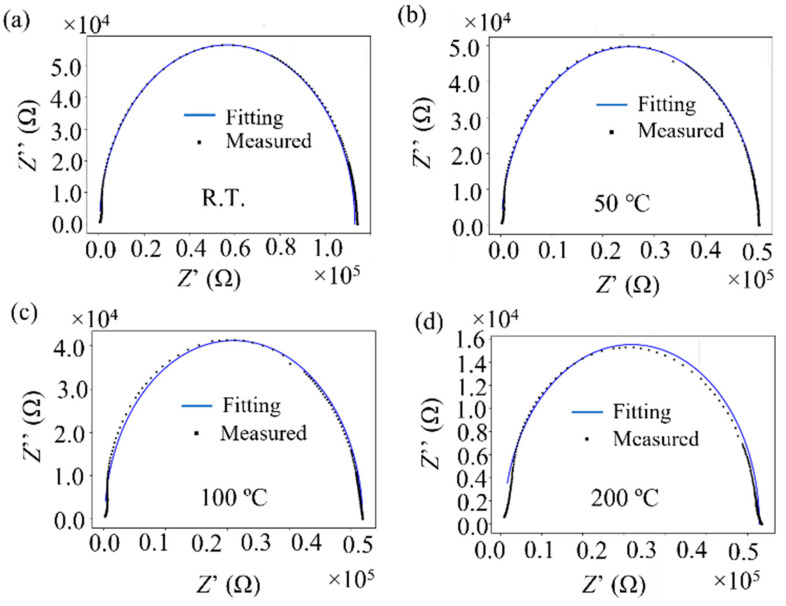
Fitting of Equation (1) to the Nyquist diagrams obtained at different temperatures: (**a**) Room temperature (23 °C); (**b**) 50 °C; (**c**) 100 °C; (**d**) 200 °C.

**Figure 20 materials-14-04662-f020:**
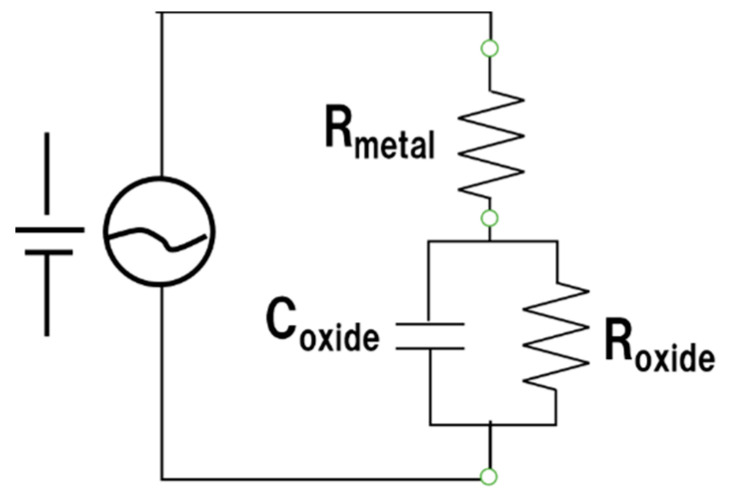
Hypothetical equivalent circuit of Inconel 718 alloy PA powder.

**Figure 21 materials-14-04662-f021:**
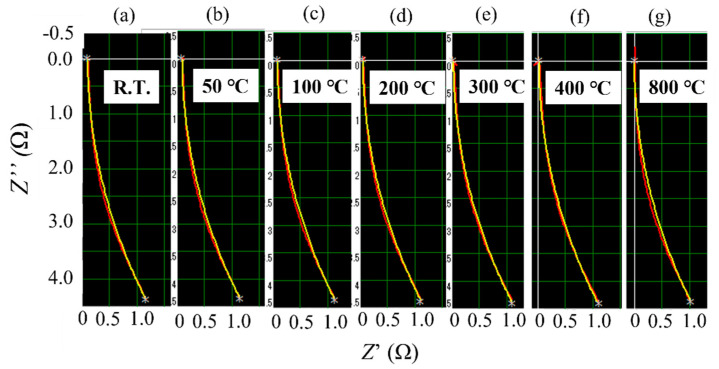
Experimental (yellow curve) and fitted (red curve) Nyquist plots of ball-milled (30 min) Inconel 718 alloy powder at different temperatures: (**a**) Room temperature (23 °C); (**b**) 50 °C; (**c**) 100 °C; (**d**) 200 °C; (**e**) 300 °C; (**f**) 400 °C; (**g**) 800 °C.

**Figure 22 materials-14-04662-f022:**
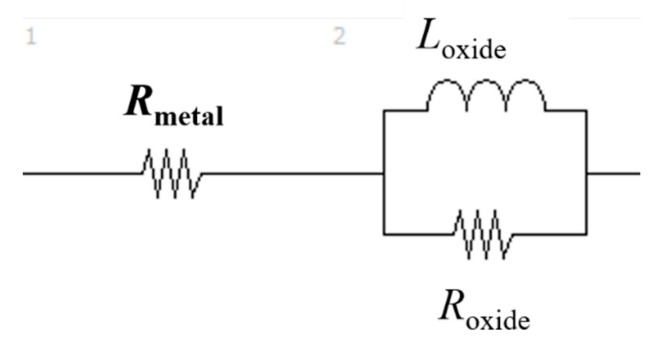
Hypothetical equivalent circuit of ball-milled Inconel 718 alloy PA powder at temperatures between room temperature and 800 °C.

**Figure 23 materials-14-04662-f023:**
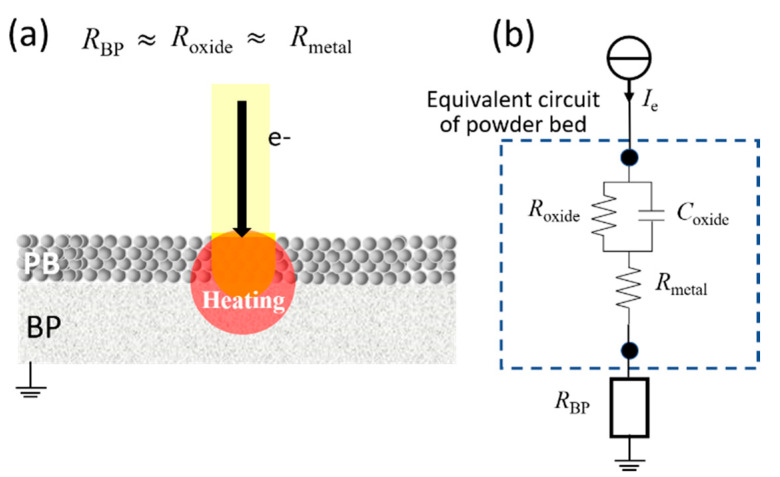
Schematic of powder bed (PB) irradiation process and equivalent circuit: (**a**) Electron beam irradiation of PB on a base plate (BP); (**b**) Equivalent circuit of electron beam/PB/BP system during electron beam irradiation. *R*_BP_, *R*_oxide_, and *R*_metal_ are the electrical resistances of the BP, oxide layer, and metal bulk, respectively.

**Table 1 materials-14-04662-t001:** Chemical composition of plasma-atomized (PA) Inconel 718 alloy powder and corresponding specification UNS N07718 (in wt.%).

Composition	Ni	Cr	Mo	Al	Ti	Nb	Mn	C	N	Fe
PA	52.48	18.80	3.03	0.50	0.92	5.10	0.07	0.044	0.022	Bal.
UNS N07718	50–55	17–21	2.80–3.30	0.2–0.8	0.65–1.15	4.75–5.50	0.35 max.	0.08 max.	---	Bal.

**Table 2 materials-14-04662-t002:** Summary of smoke tests.

Powder	*f*_1_ (Hz) ^1^	*f*_2_ (kHz) ^2^	BC (mA) ^3^	Spot Size (m) ^4^	Temperature (℃)
Untreated	100	10	20	1.8	23 (RT)
Untreated	100	10	20	1.8	600
Untreated	100	10	20	1.8	700
Untreated	100	10	20	1.8	800
Ball-milled	100	10	20	1.8	23 (RT)
Ball-milled	100	10	20	1.8	400
Ball-milled	100	10	20	1.8	500
Ball-milled	100	10	20	1.8	700

^1^ *f*_1_ = beam irradiation frequency; ^2^ *f*_2_ = reciprocal of beam dwell time (beam dwell time = 1/*f*_2_); ^3^ BC = beam current; ^4^ spot size = beam diameter.

**Table 3 materials-14-04662-t003:** Hatching (powder bed melting) test conditions.

Powder	Beam Current (mA)	Scan Velocity (mm/s)	Line Offset (mm)	Beam Spot Size (μm)
Ball-milled	1.2	200	0.2	350

**Table 4 materials-14-04662-t004:** Characteristic parameters related to particle size distribution of Inconel 718 PA powders before and after ball milling for 30 min in air.

Powder	D10 (μm)	D50 (μm)	D90 (μm)	Average (μm)
Untreated	51.6	77.0	97.9	73.5
Ball-milled	45.3	71.2	100.0	68.7

**Table 5 materials-14-04662-t005:** Summary of the results of the smoke detection tests for the virgin and ball-milled (30 min) powders, measured at *I* = 20 mA, *f*_1_ = 100 Hz, 1/*f*_2_ = 0.1 ms (*f*_2_ = 10 kHz), beam diameter (full width at half maximum; FWHM) of 1.8 mm, and preheating temperatures between room temperature and 800 °C.

Powder	Temperature (℃)	Smoke (s)
Untreated	23 (RT)	72.8
Untreated	600	497.2
Untreated	700	1400
Untreated	800	N.D.
Ball-milled	23 (RT)	N.D.
Ball-milled	400	N.D.
Ball-milled	500	N.D.
Ball-milled	700	N.D.

**Table 6 materials-14-04662-t006:** Temperature dependence of resistance and capacitance of Inconel 718 PA powder, as determined by AC impedance measurements.

Temperature (°C)	*R*_metal_ (Ω)	*R*_oxide_ (Ω)	*C*_oxide_ (F)	*τ* (*R*_oxide_*C*_oxide_) (s)	*P*
23 (RT)	477	1.13 × 10^5^	2.51 × 10^−11^	2.84 × 10^−6^	1.000
50	475	1.01 × 10^5^	2.43 × 10^−11^	2.45 × 10^−6^	0.991
100	502	8.33 × 10^4^	2.55 × 10^−11^	2.12 × 10^−6^	0.993
200	475	3.10 × 10^4^	2.96 × 10^−11^	0.918 × 10^−6^	1.000

**Table 7 materials-14-04662-t007:** Temperature dependence of the resistance and inductance of ball-milled Inconel 718 alloy PA powder.

Temperature (°C)	*R*_metal_ (Ω)	*R*_oxide_ (Ω)	*L* (H)
23 (RT)	1.18 × 10^−1^	19.2	400 × 10^−9^
50	1.09 × 10^−1^	19.2	398 × 10^−9^
100	9.66 × 10^−2^	19.2	398 × 10^−9^
200	6.68 × 10^−2^	19.2	400 × 10^−9^
300	4.42 × 10^−2^	19.2	402 × 10^−9^
400	3.33 × 10^−2^	19.2	402 × 10^−9^
800	0.73 × 10^−2^	19.7	398 × 10^−9^

## Data Availability

The data presented in this study are available on request from the corresponding author.

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
