# Peer review of "Smoke Suppression in Electron Beam Melting of Inconel 718 Alloy Powder Based on Insulator–Metal Transition of Surface Oxide Film by Mechanical Stimulation"

_materials, 2021, doi:10.3390/ma14164662_

Round 1
Reviewer 1 Report
Excellent work on EBM of Inconel 718 powder beds. Correlation of "smoke" phenomenon with electrical properties of powder beds is clear and sound.
The authors analyze the issues related with powder removal due to electrostatic forces induced by the electron beam. It has practical interest because the authors provide a methodology for avoiding this undesired phenomena and the associated defects during EBM of Inconel powders. As far as I know, The work is original. The authors have published other articles on EBM of Inconel but not on this topic.
The authors conclude that the powder conductivity can be modified by mechanical processing of the powders prior to EBM. This is critical to avoid charging during the EBM process.
I've only found a couple of mistakes:
1- In Fig. 4b the magnification bar is wrong
2- In Fig. 5, "boll" should be changed by "ball"
Author Response
1- In Fig. 4b the magnification bar is wrong
Response1: I corrected the number 200 to 20 µm in Fig.4b.
2- In Fig. 5, "boll" should be changed by "ball"
Response2: I corrected "boll" to "ball" in Fig.5
Reviewer 2 Report
The article: "Smoke suppression in electron beam melting of Inconel 718 alloy powder based on insulator/metal transition of surface oxide film by mechanical stimulation present some results" is written very well. Some results are the same as is published in the patent PCT/JP2019/010679.
The article contains some small mistakes:
- line 137 - ... "cemented carbide" ... <= better is tungsten carbide because cemented carbide represent other hard carbides too,
- lines 141 and 216 - "rpm" <= rotation speed have a unit in SI: min-1,
- in my opinion, the word distortion (distorted) was not correctly used (chapter 3.1 and Conclusion in line 716), and maybe as misshapen or classically deformed,
- the marker on Fig. 4 b is not correct 200 µm is a very high number, in the patent is value 20 µm,
- Fig. 5 contain a mistake in the legend "Boll milled",
- line 255 (and other) - "high resolution" <= better is magnification,
- some Figs. 6 ÷ 9 have a different axis description (units are in brackets) as other graphs (units are behind the slash symbol). It needs unification.
- Figs 11, 12, and 19 - use the symbol Ω in x-axis instead of "Ohm" as is in other graphs,
- correct at% to at.%;
- correct mass% to wt.%,
- lines 457, and 471 - correct mm/s to mm.s-1.
Author Response
The article contains some small mistakes:
- line 137 - ... "cemented carbide" ... <= better is tungsten carbide because cemented carbide represent other hard carbides too,
Response 1: The authors correct cemented carbide to the tungsten carbide as the reviewer’s suggestion.
- lines 141 and 216 - "rpm" <= rotation speed have a unit in SI: min-1,
Response 2: The authors correct rpm to min-1 as the reviewer’s suggestion.
- in my opinion, the word distortion (distorted) was not correctly used (chapter 3.1 and Conclusion in line 716), and maybe as misshapen or classically deformed,
Response 3: The authors change “distortion and distorted” to “deformation” as the reviewer’s suggestion.
- the marker on Fig. 4 b is not correct 200 µm is a very high number, in the patent is value 20 µm,
Response 4: Magnification number is mistaken. The authors revised 200μm to 20 μm in the SEM image.
- Fig. 5 contain a mistake in the legend "Boll milled",
Response 5: The authors corrected Boll milled to ball milled as the reviewer’s suggestion.
- line 255 (and other) - "high resolution" <= better is magnification,
Response 6: The authors changed term "high resolution" to “high magnification” as the reviewer’s suggestion.
- some Figs. 6 ÷ 9 have a different axis description (units are in brackets) as other graphs (units are behind the slash symbol). It needs unification.
Response 7: The authors unified the axis description as the reviewer’s suggestion.
- Figs 11, 12, and 19 - use the symbol Ω in x-axis instead of "Ohm" as is in other graphs,
Response 8: The authors changes "Ohm" to the symbol “Ω” as the reviewer’s suggestion.
- correct at% to at.%;
Response 9: The authors corrected at% to at.% as the reviewer’s suggestion.
- correct mass% to wt.%,
Response 10: The authors corrected mass% to wt.%, as the reviewer’s suggestion.
- lines 457, and 471 - correct mm/s to mm.s-1.
Response 11: The authors corrected correct mm/s to mm.s-1 as the reviewer’s suggestion.

Reviewer 3 Report
This investigation paper is about the EBM process treatment technology, one of the latest 3D printing technologies used in various industrial fields, and a fundamental study on the particle change in the EBM forming process using Inconel 718 material.
This paper is widely used as a good results paper for a heat-resistant alloy material considered.

Author Response
[1] Columns 126, In Table 1, the authors recorded only the ingredients of Inconel 718 PA powder.
I suggest that it is expected that readers understand better if the ingredients of the actual Inconel
718 Bulk material are additionally indicated.
Response 1: I indicated the corresponding composition of alloy In 718 as specification of UNs07718 in the Table 1.
[2] Columns 127, the author indicated (a)~(c) for Inconel 718 PA powder photo in the powder
structure photo of (a)~(c) of Figure 1.
I suggest that the (d), (e), (f) is displayed the same condition diagram as (a)~(c) same magnification.
That is easy understand for readers to understand the specimen's microstructure for this paper's
Response2: Fig. 1d,e and f show SEM images of the appearance of Inconel PA powder for 10 minutes, 30 minutes, and 60 minutes of ball milling, respectively. It is a diagram showing how the morphology of the powder changes depending on the difference in ball mill time, so it is better to show it at the same magnification. Therefore, The authors think that it is okay to keep it as it is.
[3] Columns 227, The author, please check the scale bar suitable for the magnification of the picture
(b) in Figure 4.
Could the author review that the scale bar in (b) is 20 microns compared to (b) in Figure 1?
Response3: As pointed out by the reviewers. The scale is a mistake of 20 μm instead of 200. The author correct 200 μm to 20 μm.
Reviewer 4 Report
A short abstract and an introduction in which the authors made a few editorial mistakes, relating, among others, to the method of citing the literature. Papers should contain a nomenclature.
The nomenclature should be very extensive - I believe that the abbreviations that appear in the text for the first time should also be discussed in the text, especially since they appear in the introduction. I discourage authors from using abbreviations in the abstract - they cannot be there, as well as other markings in the form of symbols - please correct the abstract and complete the introduction.
Please do not use the words "work, works" in relation to the scientific article. The words "paper, manuscript, scientific paper" should be used.
I advise against citing scientific works in the abstract – please change it.
Please systematize the font size in the figures - it must be the same everywhere, especially in charts - this applies to the descriptions of the ordinate and abscissa axes and the font for numerical values for both axes - at present it is not properly prepared. The figures must be corrected - they must be clear, and the font must be the same size everywhere.
Please make sure that the captions for figures and tables are in the right place. Currently, not all of them are where they should be.
In general, I evaluate the paper well. Proper introduction, theoretical introduction, presentation of results, correct conclusions. Please complete it with the nomenclature and send it for another review.
Author Response
A short abstract and an introduction in which the authors made a few editorial mistakes, relating, among others, to the method of citing the literature. Papers should contain a nomenclature.
- The nomenclature should be very extensive - I believe that the abbreviations that appear in the text for the first time should also be discussed in the text, especially since they appear in the introduction. I discourage authors from using abbreviations in the abstract - they cannot be there, as well as other markings in the form of symbols - please correct the abstract and complete the introduction.
Pesponse1 Thank you for your suggestion. I modified the abbreviation and nomenclature.
- Please do not use the words "work, works" in relation to the scientific article. The words "paper, manuscript, scientific paper" should be used.
Response2: We use paper instead of work in the text as the reviewer’s suggestion.
- I advise against citing scientific works in the abstract – please change it.
Response 3: We remove citing the scientific work (i.e., Mott insulator) in the abstract.
- Please systematize the font size in the figures - it must be the same everywhere, especially in charts - this applies to the descriptions of the ordinate and abscissa axes and the font for numerical values for both axes - at present it is not properly prepared. The figures must be corrected - they must be clear, and the font must be the same size everywhere.
Response 4: The authors corrected the font size and the font unification in the figures as suggested by the reviewer. We also corrected the figure ordinate and abscissa axes.
- Please make sure that the captions for figures and tables are in the right place. Currently, not all of them are where they should be.
Response 5: The authors make sure to put the figure and table in the right place of the text.
In general, I evaluate the paper well. Proper introduction, theoretical introduction, presentation of results, correct conclusions. Please complete it with the nomenclature and send it for another review.
